# "*We know about schistosomiasis but we know nothing about FGS*": A qualitative assessment of knowledge gaps about female genital schistosomiasis among communities living in *Schistosoma haematobium* endemic districts of Zanzibar and Northwestern Tanzania

**Humphrey D. Mazigo**[1]*, **Anna Samson**[2], **Valencia J. Lambert**[3], **Agnes L. Kosia**[4], **Deogratias D. Ngoma**[5], **Rachel Murphy**[6], **Dunstan J. Matungwa**[7,8]

**1** Department of Parasitology and Entomology, Weill Bugando School of Medicine, Catholic University of Health and Allied Sciences, Mwanza, Tanzania, **2** Department of Behavioral Sciences, School of Public Health, Catholic University of Health and Allied Sciences, Mwanza, Tanzania, **3** Center for Global Health, Weill Cornell Medicine, New York City, New York, United States of America, **4** School of Nursing, Catholic University of Health and Allied Sciences, Mwanza, Tanzania, **5** Accelerating the Sustainable Control and Elimination of Neglected Tropical Diseases—Crown Agents, London, United Kingdom, **6** Crown Agents, London, United Kingdom, **7** Department of Sexual and Reproductive Health, National Institute for Medical Research, Mwanza, Tanzania, **8** Department of Anthropology, School of Arts and Sciences, Rutgers University, New Brunswick, New Jersey, United States of America

* humphreymazigo@gmail.com

## Abstract

### Background

*Schistosoma haematobium* causes urogenital schistosomiasis and is widely distributed in Tanzania. In girls and women, the parasite can cause Female Genital Schistosomiasis (FGS), a gynecological manifestation of schistosomiasis that is highly neglected and over-looked by public health professionals and policy makers. This study explored community members' knowledge, attitudes and perceptions (KAP) on and health seeking behavior for FGS.

### Methods/Principal findings

Using qualitative research methods—including 40 Focus Group Discussions (FGDs) and 37 Key Informant Interviews (KIIs)—we collected data from 414 participants (Males n = 204 [49.3%] and Females n = 210 [50.7%]). The study engaged 153 participants from Zanzibar and 261 participants from northwestern Tanzania and was conducted in twelve (12) purposively selected districts (7 districts in Zanzibar and 5 districts in northwestern Tanzania). Most participants were aware of urogenital schistosomiasis. Children were reported as the most affected group and blood in urine was noted as a common symptom especially in boys. Adults were also noted as a risk group due to their involvement in activities like paddy farming that expose them to infection. Most participants lacked knowledge of FGS and

**Data Availability Statement:** The qualitative data for this study contains information that can make participants identifiable. In addition, participants did not consent to have their full transcripts made publicly available. The data policy exception related to privacy concerns pertains in this case. Data are available at the Directorate of Research and Publication, Catholic University of Health and Allied Sciences (https://www.bugando.ac.tz/) and e-mail: vc@bugando.ac.tz.

**Funding:** This work was supported by the Task Force for Global Health, Coalition for Operational Research on Neglected Tropical Diseases (NTD-SC-208.2D). H. D. M. received additional funding from UK Foreign Commonwealth and Development Office (FCDO) through the Accelerating the Sustainable Control and Elimination of Neglected Tropical Diseases (ASCEND) programme (FCDO grant number PO-8374). The funders had no role in study design, data collection and analysis, decision to publish, or preparation of the manuscript.

**Competing interests:** The authors have declared that no competing interests exist.

acknowledged having no knowledge that urogenital schistosomiasis can affect the female reproductive system. A number of misconceptions on the symptoms of FGS and how it is transmitted were noted. Adolescent girls and women presenting with FGS related symptoms were reported to be stigmatized, perceived as having a sexually transmitted infection (STI), and sometimes labeled as "prostitutes". Health seeking behavior for FGS included a combination of traditional medicine, self-treatment and modern medicine.

## Conclusion/Significance

Community members living in two very different areas of Tanzania exhibited major, similar gaps in knowledge about FGS. Our data illustrate a critical need for the national control program to integrate public health education about FGS during the implementation of school- and community-based mass drug administration (MDA) programs and the improvement of water, sanitation and hygiene (WASH) facilities.

## Author summary

Female Genital Schistosomiasis (FGS) is a disease of the female reproductive system caused by infection with *Schistosoma haematobium*, a parasite acquired through skin contact with infested water. Although this disease is highly endemic in many parts of sub-Saharan Africa and associated with serious effects on reproductive and mental health, FGS is often neglected by public health professionals and policy makers. The knowledge, attitudes, and practices (KAP) of FGS in endemic communities—important for adherence to intervention measures—are unknown. This study used qualitative research methods to explore community members' KAP on and health seeking behavior for FGS. Overall, community members knew about urogenital schistosomiasis, but lacked knowledge of FGS. Misconceptions on its cause, symptoms and modes of transmission were common. Our study highlights the need for public health education to address FGS as part of community-based mass drug administration (MDA) programs. In mainland Tanzania, as is in Zanzibar, MDA should be extended to the communities, targeting women and adolescent girls, in order to reduce the burden of FGS. For the long-term impact of these interventions, improving the supply of water, sanitation and hygiene (WASH) is an essential part of the intervention package to end both urogenital schistosomiasis and FGS.

## Introduction

The sub-Saharan Africa region carries the highest burden of schistosomiasis, accounting for 95% of the >230 million cases estimated to occur worldwide [1–3]. Approximately two-thirds of these cases are caused by *Schistosoma haematobium*, a diecious blood parasite acquired through skin contact with contaminated freshwater [1, 2]. The parasite is endemic in many communities characterized by low socio-economic status with poor or inadequate water, sanitation and hygiene (WASH) infrastructure [4]. In endemic areas, all community members, irrespective of their age and gender, are at risk of infection whenever their skin contacts with infested water [5–8]. However, specific gendered roles and norms may increase the risk of infection for some groups. For instance, girls and women of reproductive age perform household and other related activities—such as washing clothes and dishes, fetching water for

domestic use, and paddy farming—that put them at high risk as they involve requisite skin contact with water [7, 9]. In contrast, boys and men mainly contract the infection through swimming, paddy farming, washing clothes and fishing [9, 10]. In both sexes, *S. haematobium* affects the urogenital system (the urinary tract and the genital tracts) [11]. Specifically, in an estimated 40–56 million women and girls, *S. haematobium* causes a gynecological disease known as Female Genital Schistosomiasis (FGS) [12, 13].

Clinical manifestations of FGS include vaginal bleeding, vaginal itching, pain during sexual intercourse, and formation of sandy patches on the cervix and uterus [13–15]. FGS has been linked to increased susceptibility to HIV in women [16–19]. Untreated FGS can lead to infertility, miscarriage, ectopic pregnancies, and spontaneous abortions [16]. In addition, untreated FGS can lead to depression and social stigma in girls and women with the symptoms of this disease (because of being perceived to have a sexually transmitted infection [STI]) or who are struggling with infertility [20]. Yet, FGS remains highly neglected. For instance, at the time of submission, FGS had not been included as part of schistosomiasis by the Global Burden of Diseases Study [21] and is largely ignored by public health professionals, policy makers and endemic communities [21–23]. In highly endemic areas, standard diagnostic equipment, treatment, skilled healthcare workers and prevention services are inadequate [16]. Moreover, most communities lack awareness of FGS and diagnosing it is not given priority among adolescent girls and women having reproductive health issues [24].

Little is known about communities' knowledge, attitudes, and practices (KAP) on and health seeking behavior for FGS. At the time of submission, only one study in Africa had assessed the community KAP on FGS [24]. It is generally accepted that good KAP of endemic communities play a significant role in attaining sustainable disease control [25]. Moreover, design and implementation of effective behavioral change interventions to reduce risk and promote care-seeking require a clear understanding of the community's knowledge and perception of the targeted disease. In that context, this study aimed to understand communities' KAP on and health seeking behavior for FGS. We conducted the study in two parts of Tanzania with very disparate cultural and religious beliefs, but similar levels of *S. haematobium* endemicity (prevalence). In both areas, we also sought community recommendations on how best to promote community awareness of FGS and health seeking behavior among girls and women of reproductive age in all endemic areas of Tanzania.

## Methods

### Ethics statement

This study received ethical approvals from the Lake Zone Institutional Review Board (certificate number MR/53/100/649), Zanzibar Health Research Ethics Committee (certificate number ZAHREC/03/PR/DEC/2020/29), and Weill Cornell Medicine (certificate number 20–07022381). Regional and district authorities where the study was conducted granted permission to enable the study to be conducted in their jurisdiction. The study team adhered to all ethical principles. Adult participants aged 18 years and above gave a written informed consent. Participants below the age of 18 years gave a written assent and their parents or guardians gave a written informed consent for them to participate in the study. For participants who could neither read nor write, the research assistant read aloud the consent form for them, addressed their questions and concerns if they had any, and asked them to provide an oral consent/ assent. Then, an adult person—their authorized legal representative—filled out their names on the consent/assent form after which the participant signed using a thumbprint. Prior to data collection, the study team conducted sensitization meetings with responsible authorities in every study village/shehia to create awareness of the study and its data collection procedures.

The study team maintained anonymity of the study participants and confidentiality of the information provided. Individual identifiers were not used throughout the study. All participants were identified using code numbers.

## Study setting

This study was conducted between September 2020 and February 2021 in schistosomiasis-endemic districts in northwestern Tanzania and Zanzibar. In northwestern Tanzania, *S. haematobium* is endemic in communities located away from Lake Victoria shoreline (i.e. farther south and southeast of the lake) while *S. mansoni*—a parasite that causes intestinal schistosomiasis—is endemic in communities located along the lake's shoreline [26, 27]. We conducted this study in 24 purposively selected villages located in districts lying within the *S. haematobium* transmission zone: Itilima, Maswa, Misungwi, Shinyanga Rural, and Kwimba [28]. Geographically, these districts are characterized by permanent and seasonal freshwater rivers, marshes, swamps, and ponds which create a good living and breeding environment for *Bulinus* snail species—specifically *Bulinus nasutus*—which host *S. haematobium* [26]. Despite repeated rounds of mass drug administration (MDA) in these districts, some villages still experience high transmission of *S. haematobium* [26, 27]. FGS is a public health concern in these districts and one study reported an average prevalence of 5% (ranging from 0–11%) [18].

In Zanzibar, the study was conducted in 16 purposively selected shehias (pl., smallest administrative areas in Zanzibar equivalent to wards) located in 4 schistosomiasis-endemic districts [29]. The districts include North A, North B, Central, and West on Unguja Island as well as Wete, Chake Chake, and Mkoani on Pemba Island. At the time of submission, no study had been conducted to measure the burden of FGS in Zanzibar.

In sum, we purposively selected study sites located in *S. haematobium* transmission zones, particularly those that depend on and are located close to the open, permanent sources of fresh water including man-made dams, marshes, swamps, and rivers. In selecting these sites, we also depended on the historical data from the district-level neglected tropical disease (NTD) control coordinators, which indicated that these communities had the highest burden of schistosomiasis. Finally, the selected sites had implemented school- and/or community-based MDA programs using praziquantel as the main schistosomiasis control measure.

## Study design and recruitment of participants

The study used qualitative research methods to allow participants to share their perspectives in light of their lived experiences and social context [30, 31]. Using Focus Group Discussions (FGDs) [32–34] and Key Informant Interviews (KIIs) [35], we collected data from adolescent boys and girls, adult men and women, older men and women, government leaders, and opinion leaders. Inclusion criteria were age (≥15 years), permanent residence in the study village/shehia, acceptance to provide a written assent/consent, and consenting in writing to have one's opinions audio-recorded. People who did not meet these criteria were excluded while those who met them were invited to participate in the study.

The study team worked with the village chairpersons, village executive officers, and shehas (pl., leaders of shehias in Zanzibar) to purposively select FGD participants using the pre-determined inclusion criteria. FGDs were conducted with adolescent boys (15–20 years), adolescent girls (15–20 years), adult women (21–45 years), adult men (21–45 years), older men (≥46 years), older women (≥46 years), and opinion leaders (of different ages). Except for FGDs conducted with opinion leaders, which involved both male and female leaders, other FGDs involved participants of the same gender.

Key informants were purposively selected and invited to participate in the study mainly based on the position they held in the community (and other inclusion criteria). Key informants included village chairpersons, village executive officers, shehas, and opinion leaders (including sheikhs, pastors, traditional healers, and community health workers).

## Data collection and sample size

Data were collected between September 2020 and February 2021. Before starting fieldwork, field supervisors and research assistants with qualitative research experience were recruited and retrained on how to collect qualitative data using FGDs and KIIs. All field supervisors and research assistants read, became conversant with, and participated in the pre-testing of the semi-structured topic guides that would later be used to collect data. In selecting venues for discussions and interviews, the research team chose places that did not have interruptions and which ensured privacy. All FGDs and KIIs were conducted at a time and venue suggested by the research team and approved by the participants. Except for FGDs with opinion leaders that were conducted by a team of both male and female research assistants, other FGDs and KIIs were led by research assistants of the same gender as participants. Each FGD was led by three research assistants, each playing a distinct role as either a moderator, a digital audio recorder operator, or a note taker. All FGDs and KIIs were conducted in Kiswahili—a native language of both the research assistants and participants—and recorded using a digital audio recorder with the participants' written assent and/or consent. A note taker made handwritten notes of key issues that came up in the FGDs. At the end of each day of data collection, the research team held a debriefing session to review the data, address issues that had emerged during the process, and plan for the next day. The sample size for this study was guided by the concept of data saturation, that is, the researchers continued conducting FGDs and KIIs to the point where they no longer provided new information and/or generated new themes [36, 37]. In total, we conducted 46 FGDs, 30 in northwestern Tanzania and 16 in Zanzibar (Table 1) as well as 37 KIIs, 29 in northwestern Tanzania and 8 in Zanzibar (Table 2).

## Data processing and analysis

After data collection, all audio-recorded FGDs and KIIs were transcribed verbatim by a team of transcribers trained on research ethics and handling of qualitative data. The second and third authors verified the transcripts ensuring that they represented their audio versions. We coded the data transcripts using both deductive (concept-driven) and inductive (data-driven) coding approaches [38]. Transcripts were coded in Kiswahili to preserve the participants' original concepts and meanings [39] which are likely to be lost during the translation process. Before coding the transcripts, the last author prepared a codebook with the initial set of deductive codes drawing from the FGD and KII semi-structured topic guides. The second, third, and fourth authors inductively coded six (6) transcripts (2 transcripts each, 1 FGD and 1 KII) to identify new codes and themes. Thereafter, the second, third, fourth and last authors discussed and refined the deductive and inductive codes and themes to produce a refined codebook. To ensure intercoder reliability [40], all transcripts were imported on the NVivo 12 Plus software and coded by the second and third authors using the refined codebook. Lastly, themes were grouped into categories and selected quotes were chosen—and translated to English—to illustrate the major findings.

## Results

### Participants' demographic characteristics

This study engaged 414 participants (Males n = 204 [49.3%] and Females n = 210 [50.7%]). Of the 414 participants, 153 (Males n = 64 and Females n = 89) were from Zanzibar and 261

**Table 1. Study participants, FGDs conducted, and study sites**

| FGD code | Participants | Participants' sex | | Village/Shehia | District | Location in Tanzania |
|---|---|---|---|---|---|---|
| | | Male | Female | | | |
| 01 | Adolescent girls | 0 | 11 | Lagangabilili | Itilima | Northwestern Tanzania |
| 02 | Adolescent boys | 8 | 0 | Mwabulugu | | |
| 03 | Adolescent girls | 0 | 8 | Simiyu | | |
| 04 | Adolescent boys | 8 | 0 | Mhunze | | |
| 05 | Adult men | 8 | 0 | Ng'wang'wita | | |
| 06 | Adult women | 0 | 8 | Mwanunui | | |
| 07 | Adult women | 0 | 8 | Bumera | | |
| 08 | Adult men | 8 | 0 | Mwazimbi | | |
| 09 | Elder women | 0 | 8 | Mhunze | | |
| 10 | Elder men | 8 | 0 | Simiyu | | |
| 11 | Opinion leaders | 6 | 2 | Mwamtani | | |
| 12 | Opinion leaders | 6 | 2 | Itubilo | | |
| 13 | Adolescent boys | 8 | 0 | Masawe | Misungwi | |
| 14 | Adolescent girls | 0 | 8 | Masawe | | |
| 15 | Adult women | 0 | 8 | Mwajombo | | |
| 16 | Adult men | 7 | 0 | Mwajombo | | |
| 17 | Adolescent girls | 0 | 8 | Manawa | Kwimba | |
| 18 | Adolescent boys | 7 | 0 | Manawa | | |
| 19 | Adult women | 0 | 9 | Ibindo | | |
| 20 | Adult men | 6 | 0 | Ibindo | | |
| 21 | Adolescent girls | 0 | 7 | Ikwingwamanoti | Shinyanga Rural | |
| 22 | Adolescent boys | 6 | 0 | Ikwingwamanoti | | |
| 23 | Adult women | 0 | 7 | Ikwingwamanoti | | |
| 24 | Adult men | 8 | 0 | Ikwingwamanoti | | |
| 25 | Opinion leaders | 8 | 1 | Ikwingwamanoti | | |
| 26 | Adolescent girls | 0 | 6 | Mwashegeshi | Maswa | |
| 27 | Adolescent boys | 7 | 0 | Mwashegeshi | | |
| 28 | Adult women | 0 | 7 | Mwaneghele | | |
| 29 | Adult men | 6 | 0 | Mwaneghele | | |
| 30 | Opinion leaders | 6 | 3 | Mwaneghele | | |
| 31 | Adolescent girls | 0 | 8 | Mwera | West | Unguja Island, Zanzibar |
| 32 | Adult men | 11 | 0 | Kinuni | | |
| 33 | Adult women | 0 | 10 | Chaani Masingini | North A | |
| 34 | Adult men | 12 | 0 | Kandwi | | |
| 35 | Adolescent boys | 9 | 0 | Kinyasini | | |
| 36 | Adult women | 0 | 8 | Bandamaji | | |
| 37 | Adult women | 0 | 8 | Kitope | North B | |
| 38 | Adult women | 0 | 12 | Miwani | Central | |
| 39 | Adolescent boys | 8 | 0 | Chambani | Mkoani | Pemba Island, Zanzibar |
| 40 | Adult women | 0 | 9 | Wambaa | | |
| 41 | Adult men | 8 | 0 | Mtambile | | |
| 42 | Adolescent girls | 0 | 9 | Uwandani | Chake Chake | |
| 43 | Adult women | 0 | 8 | Mavungwa | | |
| 44 | Adult women | 0 | 8 | Kwale | | |
| 45 | Adult men | 9 | 0 | Wawi | | |
| 46 | Adult women | 0 | 8 | Kangagani | Wete | |
| | | **178** | **199** | | | **Total** |

**Table 2. Key informants, number of KIIs, and study sites**

| KII code | Participants | Participants' sex | | Village/Shehia | District | Location in Tanzania |
|---|---|---|---|---|---|---|
| | | Male | Female | | | |
| 01 | Sheikh | M | | Mhunze | Itilima | Northwestern Tanzania |
| 02 | Community Health Worker | | F | Mhunze | | |
| 03 | Community Health Worker | | F | Simiyu | | |
| 04 | Traditional healer | M | | Mwanunui | | |
| 05 | Village Executive Officer | M | | Ng'homango | | |
| 06 | Village chairperson | M | | Ng'wang'wita | | |
| 07 | Influential man | M | | Sagata | | |
| 08 | Retail drug seller | | F | Sagata | | |
| 09 | Influential woman | | F | Sagata | | |
| 10 | Pastor | M | | Nanga | | |
| 11 | Influential man | M | | Tagawi | | |
| 12 | Influential woman | | F | Mwamsheni | | |
| 13 | Traditional healer | M | | Masawe | Misungwi | |
| 14 | Village chairperson | M | | Masawe | | |
| 15 | Pastor | M | | Mwajombo | | |
| 16 | Community Health Worker | | F | Mwajombo | | |
| 17 | Village Executive Officer | M | | Manawa | Kwimba | |
| 18 | Retail drug seller | | F | Manawa | | |
| 19 | Influential woman | | F | Ibindo | | |
| 20 | Sheikh | M | | Ibindo | | |
| 21 | Traditional healer | M | | Ikwingwamanoti | Shinyanga Rural | |
| 22 | Village Executive Officer | M | | Ikwingwamanoti | | |
| 23 | Influential man | M | | Ikwingwamanoti | | |
| 24 | Retail drug seller | M | | Ikwingwamanoti | | |
| 25 | Community Health Worker | | F | Ikwingwamanoti | | |
| 26 | Pastor | M | | Mwashegeshi | Maswa | |
| 27 | Village chairperson | M | | Mwashegeshi | | |
| 28 | Retail drug seller | | F | Mwaneghele | | |
| 29 | Sheikh | M | | Mwashegeshi | | |
| 30 | Sheha | | F | Mwera | West | Unguja Island, Zanzibar |
| 31 | Sheha | M | | Chaani Masingini | North A | |
| 32 | Sheha | M | | Kinyasini | | |
| 33 | Sheha | M | | Kandwi | | |
| 34 | Sheha | M | | Mtambile | Mkoani | Pemba Island, Zanzibar |
| 35 | Sheha | M | | Wambaa | | |
| 36 | Sheha | M | | Chambani | | |
| 37 | Sheha | M | | Uwandani | Chake Chake | |
| | | **26** | **11** | | | **Total** |

(Males n = 140 and Females n = 121) from northwestern Tanzania. Table 3 summarizes the demographic characteristics of the study participants

## Emergent themes

We present the results on urogenital schistosomiasis and FGS. For urogenital schistosomiasis, our analysis came up with six interlinked themes: awareness, perceived prevalence, aetiology, symptoms, modes of transmission, and groups of people at high risk of urogenital

**Table 3. Demographic characteristics of the study participants**

| Variable | Female | Male | Total |
|---|---|---|---|
| Age groups (in years) | | | |
| 15–20 | 63 (30%) | 60 (29.4%) | 123 (100%) |
| 21–45 | 104 (49.5%) | 93 (45.5%) | 197 (100%) |
| ≥46 | 43 (20.4%) | 51 (25%) | 94 (100%) |
| **Education level** | | | |
| Did not complete primary education | 22 (10.5%) | 6 (2.9%) | 28 (100%) |
| Primary | 79 (37.6%) | 90 (44.1%) | 169 (100%) |
| Secondary | 98 (46.6%) | 93 (45.5%) | 191 (100%) |
| College | 11 (5.2%) | 15 (7.4%) | 26 (100%) |
| **Marital status** | | | |
| Single | 82 (39.1%) | 70 (34.3%) | 152 (100%) |
| Widow | 28 (13.3%) | 0 (0.0%) | 28 (100%) |
| Married | 100 (47.6%) | 134 (65.6%) | 234 (100%) |
| **Total** | **210** | **204** | **414** |

schistosomiasis infection. For FGS, our analysis came up with eight interlinked themes: awareness, aetiology and transmission, symptoms, women at high risk of infection, associating FGS with other diseases or medical conditions, community perception of FGS-infected women and girls, treatment seeking behavior for FGS, and interventions to mitigate the transmission of FGS.

## Awareness of urogenital schistosomiasis

Before delving into FGS, we explored community members' awareness of urogenital schistosomiasis. The majority of the participants had heard about urogenital schistosomiasis and blood in urine was reported as a common symptom, especially among boys. Urogenital schistosomiasis was considered to be very prevalent in northwestern Tanzania. In Zanzibar, it was ranked among the top three prevalent diseases, the other two being Bacteria Vaginosis (BV) and Urinary Tract Infection (UTI). Although children were considered to be at higher risk of contracting the disease than any other group, women and men were equally mentioned to be at risk because of engaging in activities such as paddy farming that involve skin contact with infested water. Transmission of schistosomiasis was often reported to be associated with skin contact with infested water. Misconceptions including that urogenital schistosomiasis was sexually transmitted were noted in both mainland Tanzania and Zanzibar. Participants did not clearly understand the difference between aetiology (cause) and the modes of transmission of FGS. Major community perspectives on urogenital schistosomiasis in mainland Tanzania and Zanzibar are summarized in Table 4.

## Awareness of Female Genital Schistosomiasis

Most participants in both parts of Tanzania had never heard of FGS despite knowing about urogenital schistosomiasis, its urogenital symptoms, and risk factors for acquisition. Despite the same parasite typically causing both urinary and genital symptoms, perspectives were summarized well by a female respondent's admission that "*I have never heard of it [schistosomiasis] in the female genitalia. I have heard of it in the bladder.*" (FGD 07, Adult women, Itilima, Northwestern Tanzania). A key informant in Zanzibar acknowledged that "*I have never heard about FGS.*" (KII 33, Sheha, North A, Unguja Island, Zanzibar). Despite having never heard

**Table 4. Community members' knowledge and perceptions about urogenital schistosomiasis in Northwestern Tanzania and Zanzibar**

| Themes | Summary of the participants' views | Illustrative Quotations |
|---|---|---|
| Awareness of urogenital schistosomiasis | Most participants were aware of urogenital schistosomiasis. Participants' sources of information about schistosomiasis included school, home, and the community. | • "*I have heard about it at school, at home, and generally from our community.*" (FGD 02, Adolescent boys, Itilima, Northwestern Tanzania). |
| | Some participants were aware of schistosomiasis because they had been infected by it at some point in their lives. | • "*I once tested [. . .] and was diagnosed with schistosomiasis.*" (KII 01, Sheikh, Itilima, Northwestern Tanzania).<br>• "*Yeah, I contracted schistosomiasis when I was still young, at the age of six, studying in primary school. I was infected and treated at home by my parents using herbal medicine. [Even then] Schistosomiasis was well known [by many people. In Kisukuma language, we call it* kisambale. *It is a disease that has existed for many years.*" (FGD 25, Opinion leaders, Shinyanga Rural, Northwestern Tanzania). |
| Symptoms of urogenital schistosomiasis | Most participants were aware of most of the chronic symptoms of urogenital schistosomiasis including abdominal/pelvic pain, difficulty passing urine, pain during urination, frequent urination, and blood in urine (haematuria). | • "*When you are suffering from schistosomiasis, you start experiencing pain. At the end of urination, blood comes out. When you see a heavy bloody discharge, it is a sign that the infection has become chronic.*" (KII 35, Sheha, Mkoani, Pemba Island, Zanzibar).<br>• "*Another symptom of schistosomiasis is abdominal pain.*" (FGD 25, Opinion leaders, Shinyanga Rural, Northwestern Tanzania).<br>• "*When you urinate, your urine is hot, it comes in small amounts, and it hurts [pain during urination]. Those are some of the symptoms of schistosomiasis.*" (FGD 06, Adult women, Itilima, Northwestern Tanzania).<br>• "*The stomach hurts a lot [abdominal pain]. And he/she [the infected person] urinates frequently. When peeing, there comes blood at the end.*" (FGD 05, Adult men, Itilima, Northwestern Tanzania).<br>• "*Some of the symptoms that a person suffering from schistosomiasis [. . .] shows are feeling severe pain during urination, drops of blood at the end of urination, and abdominal pain.*" (FGD 18, Adolescent boys, Kwimba, Northwestern Tanzania).<br>• "*If you are suffering from schistosomiasis, you will definitely know that this is schistosomiasis. Your urine changes and blood appear in your urine at the end of urination. So, you will just know it is schistosomiasis.*" (FGD 34, Adult men, North A, Unguja Island, Zanzibar). |
| | A few participants were aware of some of the common symptoms of urogenital schistosomiasis including muscle aches. | • "*When someone is infected with parasitic worms that can cause schistosomiasis, they do not feel well. They experience muscle aches. They also experience difficult urination. At the end of urination, blood comes out.*" (FGD 34, Adult men, North A, Unguja Island, Zanzibar). |
| Perceived prevalence of urogenital schistosomiasis | Most participants perceived that the infection rate of schistosomiasis was very high. | • "*In my shehia, schistosomiasis is a serious problem.*" (KII 35, Sheha, Mkoani, Pemba Island, Zanzibar).<br>• "*The information we receive from our experts [. . .] tells us that schistosomiasis is still a problem. Topographically, our places have many valleys and basins where water accumulates. Our shehia has many freshwater sources. According to experts, freshwater sources are the main source of schistosomiasis. Given this nature of the environment [surrounded by freshwater rivers], schistosomiasis is an ever-present disease in our community.*" (KII 34, Sheha, Mkoani, Pemba Island, Zanzibar).<br>• "*Schistosomiasis infection rate is so high because people use and trample on contaminated water [. . .]. When [people] go to the health facility [for diagnosis], they find that they have contracted* kisambale *[schistosomiasis].*" (FGD 09, Elder women, Itilima, Northwestern Tanzania). |
| | A few participants observed that the prevalence of urogenital schistosomiasis has declined mainly because of school-based MDA programs. | • "*These days, fewer people urinate blood because children are treated in school.*" (KII 11, Influential man, Itilima). |
| Groups of people at high risk of urogenital schistosomiasis | Participants observed that all groups of people are risk of urogenital schistosomiasis. However, young people—both male and female—are at higher risk compared to other groups. | • "*Schistosomiasis infection rate is very high among us, young people. I can say, at around ninety percent.*" (FGD 02, Adolescent boys, Itilima, Northwestern Tanzania).<br>• "*Children are at risk [of being infected with parasitic worms that can cause schistosomiasis] because they have frequent [skin] contact with water. They are not careful. Our environment is wet but they sit anywhere. They bathe in rivers, not once, not twice, but many times in one day. Therefore, children are at higher risk [than other groups].*" (KII 34, Sheha, Mkoani, Pemba Island, Zanzibar).<br>• "*I can say that schistosomiasis is a problem because it is a common disease, especially among children. Children are affected more than adults because they like playing in the ponds. [. . .] and pond water is more likely to have parasitic worms that can cause schistosomiasis.*" (FGD 27, Adult men, Maswa, Northwestern Tanzania).<br>• "*Schistosomiasis affects young people the most. [Young people] aged between fifteen and twenty years are very much affected. I think that their blood is prone to this disease. I am saying this because schistosomiasis troubled us [older men and women] too when we were young.*" (KII 22, Village Executive Officer, Shinyanga Rural, Northwestern Tanzania). |
| | Although boys and girls are both at high risk, girls have a relatively lower risk of infection than boys. | • "*Girls are at a relatively lower risk than boys because they are not free to move around in the community. Boys have a higher risk than girls because they come into contact with contaminated water frequently.*" (KII 34, Sheha, Mkoani, Pemba Island, Zanzibar). |
| | Women are at risk because of doing domestic chores which involve skin contact with contaminated water. | • "*Women are at risk because of the lack of clean and safe tap water [. . .]. Women are responsible for fetching water for household use and they carry out most of the domestic chores. We are also responsible for preparing water for our husbands [e.g. water for taking a shower]. It is expected that we make sure they get water. But where do we get it? Other groups at risk are paddy farmers and vegetable growers because they fetch water for irrigating their farms from the rivers.*" (KII 30, Sheha, West, Unguja Island, Zanzibar). |

(*Continued*)

**Table 4.** (Continued)

| Themes | Summary of the participants' views | Illustrative Quotations |
|---|---|---|
| Aetiology of urogenital schistosomiasis | A few participants were able to explain that urogenital schistosomiasis is caused by parasitic worms. | • "*Based on my understanding, schistosomiasis is a disease that is caused by parasitic worms. Other people say [it is caused by] bacteria, but we can just say they are parasitic worms. They are found in freshwater sources and particularly stagnant water. If your skin comes into contact with this water, you will be infected with parasitic worms [that can cause FGS].*" (KII 33, Sheha, North A, Unguja Island, Zanzibar). |
| | Most participants confused the aetiology of urogenital schistosomiasis (freshwater parasitic worms) with its modes of transmission (skin contact with contaminated freshwater). | • "*The main source of schistosomiasis is water. That is the source which has germs that cause schistosomiasis.*" (FGD 43, Adult women, Chake Chake, Pemba Island, Zanzibar).<br>• "*[Schistosomiasis] is caused by playing with contaminated water frequently and bathing in the river or stream water flowing through the valley. [It is caused] by taking a bath in dirty water. We should tell our children to stop playing in dirty water because that is the cause of schistosomiasis.*" (FGD 36, Adult women, North A, Unguja Island, Zanzibar). |
| Modes of transmission of urogenital schistosomiasis | Most participants understood that the transmission of schistosomiasis occurs when people suffering from schistosomiasis contaminate freshwater sources with their excreta containing parasite eggs (which then hatch in water). | • "*If a person with schistosomiasis urinates in the river and my skin comes into contact with contaminated water, I will suffer from schistosomiasis.*" (FGD 39, Adolescent boys, Mkoani, Pemba Island, Zanzibar).<br>• "*One of the people's behavior here is that they urinate in rivers. Then, some of us go to the same rivers to take a bath and wash clothes and utensils. In addition, having multiple sexual partnerships can also lead to being infected with parasitic worms that can cause schistosomiasis.*" (FGD 32, Adult men, West, Unguja Island, Zanzibar).<br>• "*Contaminated water in the ponds may contribute [to the transmission of schistosomiasis]. An infected person may bath and then urinate in that water. If another person comes to bath in the same water, then they can be infected with parasitic worms that can cause schistosomiasis.*" (FGD 02, Adolescent boys, Itilima, Northwestern Tanzania). |
| | Most participants understood that urogenital schistosomiasis is transmitted through skin contact with contaminated freshwater. | • "*Schistosomiasis is a disease that is caused by [skin contact with] contaminated water from the rivers. It is a disease that causes pain during urination and blood appears in the urine at the end.*" (FGD 33, Adult women, North A, Unguja Island, Zanzibar). |
| | Skin contact with contaminated freshwater may occur while farming in wet areas such as paddy fields. | • "*You can be infected, for instance, when farming in the paddy fields where your skin comes into contact with the already infested water flowing in from distant places and possibly passing through unsanitary places...*" (FGD 06, Adult women, Itilima, Northwestern Tanzania).<br>• "*Farming in the majaruba (paddy fields) is a risk factor. If the paddy field has those parasitic worms [causing schistosomiasis] and the person is working in there, he can be infected.*" (KII 07, Influential man, Itilima, Northwestern Tanzania).<br>• "*People who like swimming in pond water can get infected with parasitic worms that can cause schistosomiasis. Also, in this community, people are farmers and they get infected with parasitic worms that can cause schistosomiasis when they participate in agricultural activities in the wetland. When a person stands in the wetland, the parasitic worm that causes schistosomiasis enters through their skin and continues to other parts of the body.*" (KII 17, Village Executive Officer, Kwimba, Northwestern Tanzania). |
| | Skin contact with contaminated freshwater may occur when a person baths in pond water. | • "*Schistosomiasis is caused by parasitic worms that live in pond water. When you bathe in this water, you get infected with these parasitic worms.*" (FGD 25, Opinion leaders, Shinyanga Rural, Northwestern Tanzania). |
| | Skin contact with contaminated freshwater may occur because of swimming or bathing in rivers or ponds. | • "*Children are children. It is not possible for them to stop swimming in the rivers whenever they try to. Swimming in the rivers is their custom.*" (KII 34, Sheha, Mkoani, Pemba Island, Zanzibar).<br>• "*Oftentimes, children like bathing in the ponds. And they play with mud. They play in dirty places because they are not aware that playing in such places puts them at risk of being infected with diseases [such as schistosomiasis].*" (FGD 02, Adolescent boys, Itilima, Northwestern Tanzania). |
| | Some participants misconceived that schistosomiasis is transmitted through sexual intercourse. | • "*At times, you can transmit schistosomiasis to another person through sexual intercourse [...]. If I am infected with a parasitic worm that can cause schistosomiasis and then I have sex with another person, [I can transmit it to them].*" (FGD 10, Adult men, Itilima, Northwestern Tanzania).<br>• "*I have the same ideas as my fellow participant [...]. Like he said, schistosomiasis is transmitted through sexual intercourse.*" (FGD 16, Adult men, Misungwi, Northwestern Tanzania). |
| | Some participants misconceive that schistosomiasis is transmitted by sharing undergarments with an infected person. | • "*Besides having sex with an infected person, people also speculate that sharing clothes, particularly undergarments, with an infected person can transmit schistosomiasis.*" (FGD 12, Opinion leaders, Itilima, Northwestern Tanzania). |
| | Some participants misconceived that consuming a lot of salt causes schistosomiasis. | • "*My father was telling me about my grandfather who died [long ago] that he [the grandfather] contracted schistosomiasis because he was eating much salt.*" (FGD 26, Adolescent girls, Maswa, Northwestern Tanzania).<br>• "*[You get infected with parasitic worms that can cause] schistosomiasis when you eat too much salt. You know there are people who eat salt like sugar [...]. When you eat salt frequently, you just eat salt every day [...], you will be infected with parasitic worms that can cause schistosomiasis.*" (FGD 03, Adolescent girls, Itilima, Northwestern Tanzania). |

about FGS, another respondent suggested that "*if that disease [urogenital schistosomiasis] stays in the body for a long time, it will evolve into another disease [. . .] and can spread into a woman's uterus.*" (FGD 10, Adult men, Itilima, Northwestern Tanzania).

A small number of participants mentioned hearing about schistosomiasis causing infertility but this was perceived as a rare event, with one woman qualifying her statement about this by saying that "*I heard this from people, not experts. I cannot lie, I have never heard this [FGS] from experts.*" (FGD 07, Adult women, Itilima, Northwestern Tanzania).

## Aetiology and modes of transmission of Female Genital Schistosomiasis

Most of the study participants reported that they neither knew the aetiology of FGS nor its modes of transmission. As one of the female participants plainly acknowledged, "*I do not know at all. I do not know what causes FGS.*" (FGD 07, Adult women, Itilima, Northwestern Tanzania). There was confusion between aetiology and the mode of transmission. Participants related the transmission of FGS with skin contact with infested water.

"*A woman can also be infected with parasitic worms [that can cause schistosomiasis] by bathing in the river or when washing clothes in the river while standing in water. This is the main reason for getting those germs. And, women are more infected than men because the anatomy of women's reproductive system is different from that of men. This is a reason why women are more likely to be infected than men.*" (FGD 32, Adult men, West, Unguja Island, Zanzibar).

Despite relating FGS transmission with skin contact with infested water, they were not sure how urogenital schistosomiasis—which they were aware of—related to FGS. Other participants related the proximity of the urogenital bladder and the female reproductive tract as a source of infection for the female genitalia.

"*If the urinary bladder is affected, how can the vagina be spared while both are in the same area? How? The germs [that can cause FGS] spread. They do not enter and just stay in the urinary bladder or any other area where the urine passes. They spread [to other parts or organs]. The reproductive tract is close to the urinary bladder. They are neighbours. So, if you urinate blood and the organ you use is located in the same area, then germs will spread to the reproductive tract.*" (FGD 38, Adult women, Central, Unguja Island, Zanzibar).

"*You know the anatomy of a woman, right? When the parasitic worms enter the woman's [external genital organs], it is like they have entered the supermarket. They quickly spread to other parts. They can go to the reproductive tract and even the urogenital tract.*" (KII 23, Influential man, Shinyanga Rural, Northwestern Tanzania).

Others guessed what they thought could be the causes of FGS and how it could be transmitted from one person to another. Most participants thought that FGS can be transmitted through sexual intercourse either from an infected male to a female or vice versa.

"*FGS can be caused by sexual intercourse. [This is] because during sexual intercourse, if you [a man] have parasitic worms [that can cause FGS] you can leave them in a woman. Those parasitic worms will then infect her.*" (FGD 10, Adult men, Itilima, Northwestern Tanzania).

"*I think as we said earlier, [FGS] can be spread even through sexual intercourse. So, I also think that even a woman can get this infection in her reproductive tract by engaging in unprotected sex with someone [male] who is infected with parasitic worms that can cause schistosomiasis.*" (FGD 13, Adolescent boys, Misungwi, Northwestern Tanzania).

"*When a woman is infected with parasitic worms [that can cause schistosomiasis] in her genital tract, she can infect her sexual partner or her husband. During sexual intercourse, she can infect him because those bacteria stay in the genital tract.*" (FGD 04, Adolescent boys, Itilima, Northwestern Tanzania).

"*We can say that most adults get infected with parasitic worms [that can cause schistosomiasis] through sexual intercourse. That is the main source. If a woman is infected and the man is not, when they have sex, then the man will be infected.*" (KII 21, Traditional healer, Shinyanga Rural, Northwestern Tanzania).

Others reported that FGS could be transmitted by coming into contact with vapor emitted from urine or other body fluids from a person suffering from the disease.

"*A woman can be infected through urine in the toilet while urinating. As she urinates, if someone else with a similar problem [infected with parasitic worms that can cause FGS] urinated there, then there is vapor that normally comes out. That vapor enters the female genitals directly. This is one way that may contribute to being infected with parasitic worms that can cause FGS.*" (FGD 02, Adolescent boys, Itilima, Northwestern Tanzania).

"*An infected woman can infect another person when, for instance, she sits on a place and then another person comes to sit on the same place later. The infected woman can leave some tiny droplets on that place. When an uninfected person sits on the same place, the bacteria [parasitic worms that can cause FGS] are transmitted through the droplets into his/her body.*" (FGD 05, Adult men, Itilima, Northwestern Tanzania).

## Symptoms of Female Genital Schistosomiasis disease

When asked about the symptoms of FGS, most of the participants admitted that they did not know about them. Most of those who attempted to respond to this question linked the symptoms of urogenital schistosomiasis with symptoms of FGS.

"*I think someone experiences severe pain especially lower back pain. Sometimes they feel headache.*" (FGD 06, Adult women, Itilima, Northwestern Tanzania).

"*Feeling pain in private parts, blood in urine, and pain during urination. I think these are the symptoms [of FGS].*" (FGD 32, Adult men, West, Unguja Island, Zanzibar).

"*What I know is that when a woman [who is infected with parasitic worms that can cause FGS] urinates, blood will appear in urine at the end of urination.*" (FGD 08, Adult men, Itilima, Northwestern Tanzania).

Some men appeared confident when explaining the symptoms of FGS. One man described "*frequent severe abdominal pain and bright red period blood*" (FGD 08, Adult men, Itilima, Northwestern Tanzania) as the main symptoms of FGS. Another man explained the progression of FGS symptoms by saying, "*First a woman feels pain. Second, a woman fails to perform sexual intercourse. Third, the woman's genital organs hurt.*" (FGD 05, Adult men, Itilima, Northwestern Tanzania).

Symptoms that most respondents mentioned frequently included abdominal pain, pain during sexual intercourse, pain during urination, blood in urine (haematuria), miscarriage and irregular menstruation. Other symptoms that were mentioned, albeit a few times, included itching during urination, vaginal pain, fatigue, pain during sexual intercourse, frequent

urination (polyuria), loss of libido (sex drive), headache, burning sensation, infertility, sterility, severe pain before menstruation, uterine damage, leg pain, vaginal discharge (sometimes containing pus), pus in urine (pyuria), and vaginal sores.

Although most of the symptoms mentioned were indeed of FGS, it became clear that most participants guessed or derived their responses on their knowledge of (the symptoms of) urogenital schistosomiasis, intestinal schistosomiasis and sexually transmitted infections (STIs) such as gonorrhea and syphilis. This became clear particularly when respondents were probed to explain more about symptoms of FGS that they had mentioned. For example, when asked to explain what causes irregular menstruation, participants said it could be because of stomachache, urogenital schistosomiasis, frequent sexual intercourse, traumatic events, menopause, and change of environment.

## Women at high risk of Female Genital Schistosomiasis disease

Most participants perceived every female person in their communities as being at risk of being infected with parasitic worms that can cause FGS. As one female participant noted, "*This disease can infect every woman regardless of age or age group. It can infect even the elderly.*" (FGD 07, Adult women, Itilima, Northwestern Tanzania). However, girls and women of reproductive age were considered as being at higher risk than other age groups of women, followed by girls and older women. Their risk of FGS was associated with engaging in activities that involve skin contact with infested water and sexual intercourse.

"*Women and girls between fifteen and thirty-five years [are most at risk of being infected by the parasitic worms that can cause FGS]. Girls play with contaminated water in the rivers. Others [women and girls] wash clothes in contaminated water in the rivers.*" (FGD 33, Adult women, North A, Unguja Island, Zanzibar).

"*Girls of reproductive age—not like me [she was in her 40s]—go to the river, bath in contaminated water and end up being infected with parasitic worms that can cause schistosomiasis.*" (FGD 36, Adult women, North A, Unguja Island, Zanzibar).

To explain this difference in levels of risk, most respondents interestingly mentioned sexual intercourse and involvement in activities that involve skin contact with infested water (such as rice farming in paddy fields) as the two main factors that put girls and women of reproductive age at higher risk of FGS than other groups of women.

"*They are usually infected through sexual intercourse. If a woman is an adult, she will certainly engage in sexual activity.*" (FGD, 10, Adult men, Itilima, Northwestern Tanzania).

"*Those who are affected more are girls who are sexually active. Those who have multiple sexual partners. This contributes to their risk of being infected with parasitic worms that can cause FGS.*" (FGD 35, Adolescent boys, North A, Unguja Island, Zanzibar).

"*For a woman of reproductive age, parasitic worms that can cause FGS are transmitted through sexual intercourse with her husband or having sex with another partner. This is the only way she can be infected.*" (KII 09, Influential woman, Itilima, Northwestern Tanzania).

"*Women whose skin comes into contact with stagnant water, for instance, when working in the* majaruba *(paddy fields) can be infected with parasitic worms [that can cause FGS].*" (KII 06, Village chairperson, Itilima, Northwestern Tanzania).

## Associating Female Genital Schistosomiasis with other diseases or medical conditions

Many respondents admitted that they did not know how FGS was associated with other diseases or medical conditions such as HIV/AIDS and other STIs, cervical cancer, ectopic pregnancy, miscarriage, infertility and sterility. Several others made educated guesses using their knowledge of urogenital schistosomiasis or perceptions of FGS. For instance, two participants related FGS with HIV/AIDS by reasoning that:

> "*When someone has schistosomiasis, the entire urogenital system gets bruises. When she engages in sexual intercourse, she can be infected with other diseases such as HIV/AIDS.*" (FGD 05, Adult men, Itilima, Northwestern Tanzania).

> "*HIV/AIDS and schistosomiasis are different diseases according to experts. But they infect the same tract. When you get schistosomiasis, you also get HIV/AIDS. You will now have two diseases and the problem becomes severe.*" (KII 34, Sheha, Mkoani, Pemba Island, Zanzibar).

In the same topic, another participant explained the relationship between FGS and STIs and noted that a person with FGS, "*can easily get sexually transmitted infections, for example, I do not know!! Yes, is called gonorrhoea. Yes, one can get all these diseases which affect the sexual organs.*" (FGD 39, Adolescent boys, Mkoani, Pemba Island, Zanzibar).

Relating FGS and cervical cancer, a man explained that "*schistosomiasis causes wounds which can become malignant. When they become malignant, it can be a problem. [Cervical] cancer can be in the form of those wounds. This means the disease has become chronic and caused cancer.*" (FGD 05, Adult men, Itilima, Northwestern Tanzania). Other participants, however, did not know how these diseases relate. As one participant revealed, "*One can be affected by cervical cancer. But I do not have much knowledge of how this can happen. I do not have much understanding about it. I am just guessing.*" (FGD 39, Adolescent boys, Mkoani, Pemba Island, Zanzibar). Others reasoned that these two diseases did not relate. One participant reasoned that "*schistosomiasis affects the female genital tract especially the urogenital bladder and the cervix. Cancer is present in the cervix, a place where schistosomiasis does not reach.*" (FGD 04, Adolescent boys, Itilima, Northwestern Tanzania).

Other participants related FGS with infertility. One participant explained that when FGS affects "*the entire sexual reproductive system. That is when you get the problem of infertility.*" (FGD 05, Adult men, Itilima, Northwestern Tanzania). Others reasoned that FGS could cause miscarriage. One informant explained how this can happen: "*because those 'insects' [parasitic worms that can cause FGS] attack the pregnancy. When they attack the pregnancy, they lead to miscarriage.*" (KII 07, Influential man, Itilima, Northwestern Tanzania).

## Community perception of women and girls suffering from Female Genital Schistosomiasis

Participants in both parts of Tanzania, mainland and Zanzibar, reported that women and girls suffering from FGS are often labeled as "prostitutes" or unfaithful in their intimate relationships. They also reported that if a person suffering from FGS is a young girl, community members would think that she has started having sex at a young age and view her as promiscuous. Some of the participants explained these individual and community perceptions.

> "*What I know is that in our community, people will think that she [a woman or girl with the symptoms of FGS] is a prostitute.*" (KII 01, Sheikh, Itilima, Northwestern Tanzania).

"*[On observing the symptoms of FGS], the husband will not think it is [Female Genital] Schistosomiasis. He will think that she is promiscuous and that is why she got that disease. He will think that she had sex with someone suffering from a sexually transmitted infection. And they will break up*." (FGD 33, Adult women, North A, Unguja Island, Zanzibar).

"*In our village, if one [a male sexual partner] hears that their girlfriend is bleeding [showing symptoms of FGS], they will be afraid. They will suspect that she has a sexually transmitted infection and think that they will also be infected. So, they will break up [with the girl] and he will chase her away*." (FGD 14, Adolescent girls, Misungwi, Northwestern Tanzania).

However, other participants reported that as schistosomiasis, particularly urogenital schistosomiasis, was becoming better known, stigmatization was decreasing.

"*Stigmatization may occur. However, these are isolated cases because schistosomiasis is now regarded as normal problem*." (KII 07, Influential man, Itilima, Northwestern Tanzania).

Participants reported that if the infected person is a married woman, just like for girls, community members might perceive them as being promiscuous and unfaithful to their partners. Interestingly, they might also think that they have contracted STIs.

"*They will not think that this woman is suffering from schistosomiasis. They will think that she has a sexually transmitted infection*." (FGD 06, Adult women, Itilima, Northwestern Tanzania).

If the person suffering from FGS happens to be an older woman—and since older women are regarded as less sexually active—this infection will be regarded as other normal diseases. In other words, older women who suffer from FGS are not stigmatized.

"*If an older woman has those symptoms [of FGS], the community cannot claim that she has been infected by another person. They will simply say she got infected [because of other reasons other than engaging in sexual activity]*." (FGD 06, Adult women, Itilima, Northwestern Tanzania).

## Treatment seeking behavior for Female Genital Schistosomiasis

In northwestern Tanzania, participants reported that community members often rely on traditional healers or self-treatment with natural herbs to manage their illness before going to a biomedical facility to seek diagnosis and treatment. In Zanzibar, community MDA was reported as the main source of schistosomiasis treatment. They also mentioned traditional healers and health facilities as the source of treatment.

"*In our setting, many people rush to the traditional healers. Others go to the dispensaries, health centres, and hospitals. But the decision really depends on a person's level of education [level of awareness of treatment options and efficacy]*." (FGD 05, Adult men, Itilima, Northwestern Tanzania).

"*Some people get medication over the counter. They buy it or get it for free from another person and treat themselves [without diagnosis]*." (KII 07, Influential man, Itilima, Northwestern Tanzania).

"*The government is aware of this disease [schistosomiasis]. Therefore, it provides drugs in different areas. There are distributors in our community. People who feel that they are infected can go and get the drugs*." (FGD 35, Adolescent boys, North A, Unguja Island, Zanzibar).

"*I do not know where the drugs [for schistosomiasis] are sold. I cannot afford to buy drugs. But we are used to getting them from our Sheha.*" (FGD 39, Adolescent boys, Mkoani, Pemba Island, Zanzibar).

### Interventions to mitigate the transmission of Female Genital Schistosomiasis

We asked study participants to suggest the best interventions that could be implemented at different levels to mitigate the transmission of FGS. We categorized their suggestions into two groups: (1) interventions aimed at improving the availability of FGS services for women suffering from this disease, and (2) interventions aimed at integrating FGS, Sexual and Reproductive Health (SRH), STIs and cervical cancer screening services.

### Improving the availability of Female Genital Schistosomiasis services

Participants suggested that in order to mitigate the suffering from undiagnosed and untreated FGS among women and girls, the government—either alone or with support from Non-governmental Organizations (NGOs)—needs to improve the availability and delivery of FGS services. They suggested three forms of interventions. First, they suggested increasing diagnostic capacity for FGS at local dispensaries and health centres, making this care accessible to patients who currently must travel long distances to seek these services at the hospitals.

"*In our place, diagnostic services are not available. Health facilities that have laboratories are far from here. If you want to do a checkup today, you have to travel far. And because the economy is not good, even if you do not feel well, you will keep postponing to go. But if the services are located near to us, if you feel [you are suffering from FGS], you will go for the services.*" (KII 01, Sheikh, Itilima, Northwestern Tanzania).

Second, they suggested that authorities should recruit more health workers. They also suggested that the government should train more health workers so that they can have skills to provide services in a friendly manner.

"*The government should employ more health workers. For instance, we have few doctors and nurses here. If we have enough health workers, people will get good health care and services within a short time.*" (FGD 26, Adolescent girls, Maswa, Northwestern Tanzania).

"*The government should plan to train [and equip] health workers with skills to provide counseling [and other services] professionally. Not these rude [service providers]. Sometimes you come across a mean service provider. Maybe they are in a bad mood that day. You approach them for a service that they know is available [at their health facility], but they refer you to Nyambiti hospital [located far from this village] where the service is more expensive.*" (FGD 18, Adolescent boys, Kwimba, Northwestern Tanzania).

Third, they suggested the need to ensure medications are available for patients to access and use when they go to the health facilities.

"*The other thing is that you find that the medications are out of stock. I do not understand how the medications get out of stock. Maybe they provide services to many people or sell them? I cannot prove this. But we usually face the shortage of essential medications.*" (KII 10, Pastor, Itilima, Northwestern Tanzania).

"*The government should ensure medications are available. The stock should be increased so that doctors can prescribe medications to their patients here [at the health centre] instead of referring them to Maswa [District Hospital]. For example, if a pregnant woman comes here, they should be able provide her with all services, not referring her to Maswa [District Hospital].*" (FGD 26, Adolescent girls, Maswa, Northwestern Tanzania).

## Integrating female genital schistosomiasis services, sexual reproductive health, sexually transmitted infections and cervical cancer screening services

Participants suggested that creating awareness of FGS, cervical cancer, and HIV was the key intervention that would promote women's SRH. Participants suggested a number of ways in which health education would improve women's awareness of the need for, and access to these services. First, they suggested that people need to be educated about FGS including its symptoms, and effects so that they can make informed decisions about health care seeking whenever they sense its symptoms.

"*To be motivated [to access FGS services], people should be educated about FGS, its symptoms, and effects. If you know these aspects, you will sense that may be with these symptoms, I have this or that problem [FGS] and that if I do not act [go to the hospital or any other health facility], I will end up with certain effects.*" (FGD 11, Opinion leaders, Itilima, Northwestern Tanzania).

"*The main intervention that can eliminate schistosomiasis is education. The community should be educated to adhere to interventions so that schistosomiasis can be eliminated. Here in Mwera, only drug distributors are educated. Other community members are not. It is good to educate all people so that they can know what schistosomiasis is.*" (FGD 31, Adolescent girls, West, Unguja Island, Zanzibar).

"*People should be sensitized that this disease is transmitted in this way. And that if you see such symptoms, you rush to the healthcare facility where you will be given certain specific types of treatment.*" (FGD 19, Adult women, Kwimba, Northwestern Tanzania).

Connected to this, participants also suggested that authorities can organize meetings where women and girls can be educated by health professionals about the variety of threats to women's SRH.

"*What should be done is to call a meeting and educate them [women and girls] about the problem [SRH problem] and its effects, and to encourage them to visit the health facility to find out what the problem is if they have certain symptoms.*" (FGD 02, Adolescent boys, Itilima, Northwestern Tanzania).

Others suggested that health workers or community health workers can conduct house-to-house sensitization meetings to educate people about SRH issues.

"*When you gather people together [like in a meeting], they tend to conceal their problems. They do not want everyone to know about their health problems. So, the best way is to visit every household.*" (FGD 08, Adult men, Itilima, Northwestern Tanzania).

Another benefit of health education would be to help people understand the symptoms, mode of transmission, and effects of both FGS and STIs as a way to fight stigma leveled against

women and girls suffering from these diseases. In the case of FGS, participants suggested that health workers should also be educated to treat patients suffering from FGS like other patients. With no or less stigmatization, these women and girls can then easily access healthcare services. As one leader explained,

> "*Knowledge will help people to become self-aware and understand the symptoms [of FGS]. Once she understands the symptoms, she can be free to share with people who will advise her on how and where to get treatment. So, there will be no stigma.*" (FGD 11, Opinion leaders, Itilima, Northwestern Tanzania).

Women also emphasized the importance of training health workers so that they treat patients suffering from FGS like other patients instead of stigmatizing them.

> "*They should treat them well like any other patients who come with other diseases. If you love someone and show her that you love her, she can feel happy and come [for services at the health facility] without any worries.*" (FGD 06, Adult women, Itilima, Northwestern Tanzania).

Others suggested that a mobile clinic would be beneficial in fighting stigma against FGS.

> "*Many women are scared of going to the health facilities for testing because someone will see them. Thus, if the services are brought directly to the village [. . .], many people will get tested.*" (FGD 06, Adult women, Itilima, Northwestern Tanzania).

## Discussion

Our data demonstrates a strong need for health education, particularly about FGS. In both parts of Tanzania, despite a broad and relatively good community knowledge of urogenital schistosomiasis, almost nothing was known about FGS. Further, women with symptoms of FGS are often stigmatized by other community members. Addressing gaps in knowledge is critical to the achievement of community participation and adherence to control measures against FGS. Moreover, community members provided practical suggestions on how health education can be provided. The observed strong similarities in the knowledge of urogenital schistosomiasis and gaps in the knowledge of FGS from communities with very different cultures suggests that our findings are broadly transferable throughout Tanzania and beyond.

FGS is highly overlooked and neglected by local, regional, and global health professionals and policy makers [12, 23, 41]. Consistently, in Tanzania, there have been very few community-based studies addressing FGS in adolescent girls and women [18, 42–44] and none has assessed the KAP of these groups on FGS. Our findings indicate that the majority of the participants knew about urogenital schistosomiasis but had never heard of FGS and were not aware that schistosomiasis may affect a woman's reproductive system. Similarly, in a previous study in Ghana, none of the community members who participated in FGDs reported having heard of FGS [24] and only 18.9% of participants agreed that urogenital schistosomiasis could cause reproductive health diseases [45]. In Egypt, it was similarly found that endemic communities had knowledge of urogenital schistosomiasis as a disease but did not believe that it could have any reproductive health effects [46].

Consistent with findings from these other studies, the majority of our study participants neither knew the cause of FGS nor its modes of transmission. Misconceptions on the modes of transmission were noted, with sexual intercourse, vapor coming from pit latrines, and

urinating on the same spot with an infected person being the most common responses. Our findings regarding the confusion surrounding the modes of transmission of FGS are confirmed by a similar study in Ghana [24]. In addition, participants struggled to list the symptoms of FGS. Though participants guessed or derived their responses from their knowledge of urogenital schistosomiasis and STIs, some of the mentioned symptoms were directly related to FGS by chance. In Ghana, though adult and adolescent girls discussed genital symptoms they experienced, they did not relate them to schistosomiasis [24].

Generally, women in the study communities were considered to be at risk of FGS. However, girls and women of reproductive age were considered to be at higher risk than older women. Girls and women of reproductive age were reported to engage in activities, which brought them into regular skin contact with contaminated water including economic activities such as paddy farming in the wetlands and domestic chores such as washing clothes in contaminated rivers and ponds. These gendered roles—which require women to regularly come into contact with cercariae-contaminated water—increase women's risk of being infected with parasitic worms that can cause schistosomiasis [7, 47]. In rural schistosomiasis-endemic communities in Africa, lack of alternative sources of water forces them to use contaminated water sources for domestic and agricultural purposes [8, 24]. To achieve sustainable control of schistosomiasis, specifically FGS, integrating MDA with WASH and public health education should be part of the development agenda. Improvement in water supply in endemic communities has been reported to decrease the risk of being infected with parasitic worms that can cause urogenital schistosomiasis [48–50]. Behavioral changes that limit exposure to parasitic worms that can cause schistosomiasis is one of the recommended preventive measures against schistosomiasis. However, without improving the availability of clean and safe water in endemic communities, it is difficult for people to change their behaviors related to water contact.

Socially, FGS symptoms in women and adolescent girls can lead to stigma and may influence health-seeking behavior negatively. Participants in both areas of Tanzania reported that women and girls presenting the symptoms of FGS or STIs are labeled as "prostitutes" within their communities. Similarly, girls with genital symptoms of FGS were accused of promiscuity and in health facilities they were likely to be referred for STI treatment [24]. Our findings and those from Ghana [24] clearly indicate that there is a need to implement public health interventions at the community and health facility levels to, first, create awareness of FGS, its symptoms/clinical signs, diagnosis, management, and prevention and, second, encourage women and girls to seek for treatment at health facilities or that it.

Community members' health seeking behavior was reported to rely on traditional healers, self-treatment using local herbs and biomedical treatment. Before seeking modern medicine, community members preferred to seek treatment from either traditional healers or self-treatment using either herbs or buy medicine from drug shops. Long queues at the health facilities were cited as a reason for opting for traditional medicine and self-treatment. Similar findings were reported in northwestern Tanzania [51, 52], Zanzibar [10] and Ghana [53]. Perceptions that urogenital schistosomiasis does not have serious negative health impacts, that it is a disease that can heal spontaneously without treatment, and that it is a normal disease, all led to community members opting for traditional medicine as the treatment option [51, 52]. In Ghana, adolescent girls cited stigmatization as a barrier to seeking care from health facilities and opted for home-based remedies or self-treatment [24]. Previous studies showed that women who experienced symptoms such as bleeding after sexual intercourse were reluctant to seek for treatment because of the fear of being associated with having an STI [54]. Partly, this could be related to lack of knowledge of FGS among healthcare providers. Actually, one of the major gaps in addressing FGS is inadequate capacity of healthcare providers to effectively identify, refer, and manage (diagnose and treat) women and girls with FGS [23].

Our study participants suggested practical strategies that are likely to be effective in controlling FGS in their communities, in addition to MDA using praziquantel and improvement of WASH [48, 50, 55]. The following were recommended: (1) improve availability of diagnostic services at the level of dispensaries and health centres, (2) increase the number of skilled healthcare providers who are able to offer services in a friendly manner, (3) improve availability and accessibility of praziquantel, (4) offer health education to motivate adolescent girls and women to seek care, and (5) provide in-service training to healthcare providers so they can offer FGS services without stigmatizing women and girls who present with genital symptoms. Furthermore, participants recommended integrating FGS services with other health services offered in SRH such as prevention of STIs and cervical cancer screening. Similar recommendations were provided by international, global health professionals [22].

The findings of this study revealed that the surveyed communities had adequate knowledge of urogenital schistosomiasis. For example, transmission was well known to be associated skin contact with infested water and blood in urine was the commonly described symptom. High risk activities—including those that involve skin contact with infested water—among adults and children were mostly known. However, misconceptions were also common, with some people reporting schistosomiasis as an STI. Cumulatively, the gaps of knowledge of schistosomiasis observed in the this study communities have also been observed by previous studies conducted in urogenital schistosomiasis-endemic areas in Tanzania [10, 24, 51, 52]. In Magu and Shinyanga districts, northwestern Tanzania [51, 52], Zanzibar [10] and Ghana [24, 45], urogenital schistosomiasis was considered as an STI.

Previous studies in Zanzibar [10] and northwestern Tanzania [52] noted that community members associated urogenital schistosomiasis with boys more than girls. These beliefs were also present in our study. Biomedically, urogenital schistosomiasis is highly prevalent among young populations, with children aged 8–15 years carrying the highest burden of the disease [11]. In relation to gender, conflicting evidence exist [47]. Some studies have reported that females have the highest risk and carry the highest burden of the disease while other studies have reported the opposite [8, 47]. In parasitological surveys, blood in urine detected by either visual or urine reagent dipstick is more often reported in boys than in girls [56, 57]. The relationship between schistosomiasis and boys has been repeatedly reported in endemic countries [24, 52]. Other endemic communities consider the infection (blood in urine) as normal in boys and believe that it does not have any health impact [10]. These results define the difference in exposure to risk environment between females and males when carrying out different activities in contaminated water sources [8]. Confirming results from the FGDs and KIIs, parasitological surveys have previously reported that the selected districts differ in the prevalence levels of urogenital schistosomiasis—ranging from very high levels in northwestern Tanzania [26] and very low in Zanzibar [58, 59]—and transmission occurs seasonally [29, 60]. High transmission occurs between March and May, the period of the long rainy season, and lower transmission occurs during the rest of the year [60]. In mainland Tanzania, control of schistosomiasis using praziquantel targets mainly school children and is offered at school [26] while in Zanzibar the control activities target schools and communities, thereby reaching school children and community members respectively. While repeated rounds of treatment have resulted in a decline in prevalence as well as the intensity of infection and re-infection [58, 59], a single intervention will not eliminate the disease.

In spite of the relevant findings from our study, we acknowledge a number of limitations. First, the findings are based on the participants' narratives and experiences and may thus represent skewed perspectives by those willing to participate in the discussions. Second, we did not conduct parasitological and gynecological examinations to ascertain the burden of urogenital schistosomiasis and FGS. Instead, we relied on previous studies that had confirmed high

prevalence of *S. haematobium* in the study communities. Third, some of the probes included in the data collection tools may have shaped the responses of the participants, especially on FGS. In spite of these limitations, our findings demonstrate inadequate knowledge of urogenital schistosomiasis and more specifically of FGS in different parts of Tanzania.

## Conclusions and recommendations

Our findings demonstrate that most communities living in known *S. haematobium* endemic areas of Tanzania have a relatively good knowledge of urogenital schistosomiasis but lack knowledge of FGS. Misconceptions on the aetiology and modes of transmission of urogenital schistosomiasis and FGS were common. Community members recognized the need for being educated about these diseases. Our data emphasize the urgent need for public health interventions to focus on improving community awareness of FGS, which in turn will reduce stigma and improve women and girls' health seeking behavior. Furthermore, integration of FGS services into existing SRH services with skilled healthcare providers will enhance the health seeking behavior of women and girls. Managing FGS will be an important aspect of the intervention. Not only will it lead to decreased gynecological symptoms and stigma faced by women and girls but also will reduce the risk of contracting HIV and other STIs. Lastly, for the successful control of urogenital schistosomiasis and FGS, inclusion of women and girls in the annual MDA program is highly recommended to reduce the disease prevalence and related morbidities while we pursue the ultimate goal of improving the supply of WASH to achieve schistosomiasis elimination.

## Acknowledgments

The authors would like to thank the study participants for their time and responses without which this paper would not have been possible to write. We also thank the regional, district, shehia and village authorities where the study was conducted for granting the permission and providing every support to successfully conduct this study. We are grateful to Prof. Jennifer A. Downs of the Center for Global Health, Weill Cornell Medicine, New York, United States of America for her support in conceptualizing the project whose outputs include this paper. We also thank her for the reviews, comments and suggestions on the early drafts of this article. The views expressed in this publication are those of the authors and not necessarily those of the funding agencies. Its contents are solely the responsibility of the authors and do not necessarily represent the official views of the supporting offices.

## Author Contributions

**Conceptualization:** Humphrey D. Mazigo, Dunstan J. Matungwa.

**Data curation:** Humphrey D. Mazigo, Anna Samson, Valencia J. Lambert, Dunstan J. Matungwa.

**Formal analysis:** Humphrey D. Mazigo, Valencia J. Lambert, Agnes L. Kosia, Rachel Murphy, Dunstan J. Matungwa.

**Funding acquisition:** Humphrey D. Mazigo.

**Investigation:** Humphrey D. Mazigo, Valencia J. Lambert, Dunstan J. Matungwa.

**Methodology:** Humphrey D. Mazigo, Anna Samson, Dunstan J. Matungwa.

**Project administration:** Humphrey D. Mazigo, Deogratias D. Ngoma, Rachel Murphy.

**Supervision:** Humphrey D. Mazigo, Anna Samson, Agnes L. Kosia, Deogratias D. Ngoma.

**Writing – original draft:** Humphrey D. Mazigo.

**Writing – review & editing:** Humphrey D. Mazigo, Dunstan J. Matungwa.

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
