## [Decision Letter · Decision Letter 0]

18 Aug 2021

Dear Prof. Mazigo,

Thank you very much for submitting your manuscript "“We know about schistosomiasis but we know nothing about Female Genital Schistosomiasis”: Knowledge gaps about female genital schistosomiasis among communities living in Schistosoma haematobium endemic districts of Zanzibar and Northwestern Tanzania" for consideration at PLOS Neglected Tropical Diseases. As with all papers reviewed by the journal, your manuscript was reviewed by members of the editorial board and by several independent reviewers. The reviewers appreciated the attention to an important topic. Based on the reviews, we are likely to accept this manuscript for publication, providing that you modify the manuscript according to the review recommendations. 

Sincerely,

Stefanie Knopp

Associate Editor

Justin Remais

Deputy Editor

Reviewer's Responses to Questions

**Key Review Criteria Required for Acceptance?**

**Methods**

-Are the objectives of the study clearly articulated with a clear testable hypothesis stated?

-Is the study design appropriate to address the stated objectives?

-Is the population clearly described and appropriate for the hypothesis being tested?

-Is the sample size sufficient to ensure adequate power to address the hypothesis being tested?

-Were correct statistical analysis used to support conclusions?

-Are there concerns about ethical or regulatory requirements being met?

Reviewer #1: A well-conducted qualitative study using focus group discussion and key informants interviews and in which the objectives are covered adequately with a large group of informants. 

See - Editorial and Data Presentation Modifications

Reviewer #2: Abstract and author summary are clear. 

Introduction

Line 76: WGRA - is this commonly used? I have seen WRA (women of reproductive age) used more frequently.

Line 87: FGS can lead to 

Line 98 - 99: Not sure about the statement “proper KAP”… I would suggest a less judgmental word than “proper”. Also there is evidence suggesting knowledge alone does not change behaviour - but is one of many factors. 

Methods

In the ethical considerations, did you have any cases where adult literacy was a problem and if so how did you indicate assent to participate?

In the study design, why such variation in sample between the # of FGD e.g., 30 in NW Tanzania and 16 in Zanzibar? And # of IDI - 29 in NW Tanzania and 8 in Zanzibar? IT is clear that you identified more villages in NW Tanzania, but it isn’t clear how you determined how many IDI / FGD per site. 

Another question - this is quite a large sample for a qualitative study and wondering at what point you reached saturation? Did you plan to do so many IDI and FGD from the beginning? Or were any of these added until saturation was reached? Some comment about saturation would be helpful if/how it played a role in your sample size. 

Tables 2 and 3 are very clear and helpful.

Data collection 

Data collection was clearly outlined, as was data processing and analysis.

Reviewer #3: -Are the objectives of the study clearly articulated with a clear testable hypothesis stated?

The objectives of the study are clearly articulated as follows:

== this study aimed to understand communities’ KAP on and health seeking behavior for FGS.

== we also sought community recommendations on how best to promote community awareness about FGS and health seeking behavior among WGRA in Tanzania.

-Is the study design appropriate to address the stated objectives?

== Yes, this is a qualitative research study using FGDs and KIIs to address the objectives.

-Is the population clearly described and appropriate for the hypothesis being tested?

== it would be good to know some more details about the selection criteria of the study population. What were inclusion and exclusion criteria?

-Is the sample size sufficient to ensure adequate power to address the hypothesis being tested?

= While the sample size seems large for a qualitative study, it is not clear, how the sample size was determined. If by saturation, it would be good if the authors can decsribe how saturation was reached and assessed.

-Were correct statistical analysis used to support conclusions?

== No (or only some descriptive) statistics were used in this qualitative research paper, which is fine.

-Are there concerns about ethical or regulatory requirements being met?

== What was the procedure for child assenting? Was written assent provided?

**Results**

-Does the analysis presented match the analysis plan?

-Are the results clearly and completely presented?

-Are the figures (Tables, Images) of sufficient quality for clarity?

Reviewer #1: Overall results and key findings are well presented.

See - Editorial and Data Presentation Modifications

Reviewer #2: Results

Table 4 - is it possible to id potential respondents here - particularly the opinion leaders? Also line 279? Line 299? Line 374? Line 378? The terms of ‘famous man’ and ‘famous woman’ are unclear. 

Is there a typo at the bottom of Table 4 e.g., “taking a birth” ?

Line 347 - this heading seems strange- men can’t be at any risk for FGS - so women have a “higher risk” than whom? Girls? Maybe “high risk” is a better term here?

The results are comprehensive and there may be some places where less quotes could be highlighted, even though they are interesting, picking fewer illustrative quotes may be good in some circumstances and would reduce length. It would also help the reader to include whether the site is Pemba or NW Tanzania. Without stating it, the reader must refer to the table. This would also help to illustrate the similar themes that arose in both sites.

Reviewer #3: -Does the analysis presented match the analysis plan?

== No analysis plan is presented.

-Are the results clearly and completely presented?

== The results are nicely structured and presented in tables and quotes.

-Are the figures (Tables, Images) of sufficient quality for clarity?

== Table 2 and Table 3 could be moved to supporting information.

== Table 4 is huge, but is very well structured and contains important results.

== Figure 1 and 1 b are fine.

== Figure 2 and 2b are outdated (from 2017) and since the manuscript is not about schistosomiasis elimination, I suggest to delete these figures.

**Conclusions**

-Are the conclusions supported by the data presented?

-Are the limitations of analysis clearly described?

-Do the authors discuss how these data can be helpful to advance our understanding of the topic under study?

-Is public health relevance addressed?

Reviewer #1: The qualitative results support the conclusion that FGS is indeed a neglected disease manifestation of high public health relevance in major parts of rural Africa

Reviewer #2: Discussion

Given the lack of availability of Praziquantel and/or its prohibitive cost for some community members when purchased in a pharmacy, it is surprising that you did not hear more about this in your dataset. Is PZQ readily available in Tanzania to be purchased in the pharmacy? Can the authors provide a bit more context on this?

In these areas of Tanzania, is cervical cancer screening common? Did the study probe as to why respondents thought it would be good to combine FGS with HPV/ cervical cancer screening? Indeed it has been suggested in the global literature, but assume that study respondents wouldn’t have access to that information. 

Line 685 - perhaps adding a comment about ease of reinfection?

Reviewer #3: -Are the conclusions supported by the data presented?

Yes

-Are the limitations of analysis clearly described?

Yes

-Do the authors discuss how these data can be helpful to advance our understanding of the topic under study?

Yes

-Is public health relevance addressed?

Yes

**Editorial and Data Presentation Modifications?**

Reviewer #1: Some suggestions/reflections:

line 32 - Should be written Females and Males, respectively + Write n=201 and n=204, respectively

lines 127-130 - Figure 2 placed before Figure 1. The figures (maps – made for another purpose) are confusing. Suggest to prepare new maps for the specific purpose of this manuscript.

line 135 - Relatively low – in the general population, or the general female or age specific population? With such a low prevalence, how well should the population, then be expected to be knowledgeable about schistosomiasis in general and FGS specifically

line 137- In the abstract it says 4 districts of Zanzibar

lines 146-147 - Hence, a selected, and maybe not fully representative study population, being exposed to MDA programs, and as a result received information about schistosomiasis. What about populations not being targeted in the programs, probably even lower level of health literacy. That could have been interesting to assess as well.

line 150 - In the figures, stated as Figure 1a and 1b, respectively.

line 152 - Confusing – also Figure 2 contains two maps, 2a and 2b, respectively,

line 159 - In the abstract, FGD, and not FGDs, - KII likewise

line 164 - It will give more detailed information if data about education level and marital status are presented in accordance to the three age-groups (and gender), and by use of percentages

line 164 - What were the inclusion criteria, apart from age?

Any exclusion criteria?

lines 170-172 - Suggest to place table 2 and 3 as supplements – not that important data.

line 184 - Data were collected between…

line 216 - Suggest to start with presentation of demographics (Table 1).

lines 217-219 - Which terms were used in communication with the participants in the FGDs and KII.

Were those themes already identified and applied in a guide for the FGD and KII in order to ensure a semi-structured approach?

lines 219-222 - likewise

line 227 - What were the other two? HIV, STIs ? 

lines 232-233 - !!! See comment above (lines 217-222)

Table 4 (line 235-236) - A bit confusing. Theme in Table 4 seems not necessarily to be the same as themes in text ( p.12, 216-219). Also confusing what is means by Results versus Theme in table 4

line 240 - How was FGS presented to the participants in the first place ? As female genital schistomiasis, FGS or in another way in the opening question. What was the opening question ? That must quite essential for assessment of awareness. If presented in clinical terms, most local people will not be able to recognize the subject. Same goes for other clinical terms, in particular using abbreviations. 

line 214 - 216 - How is that – when there is no knowledge about the the existence of FGS? (same goes for the other FGS themes)

line 451 - Likewise, how can FGS treatment seeking behavior be assessed when there is no basic knowledge about existence of FGS?

line 718 - Jennifer A. Downs is stated as contributing author to study conceptualization, but not stated as co-author.

In general, many abbreviations/acronyms in the manuscript are used- are they all necessary?

Reviewer #2: N/A - data is presented clearly and comprehensively in the paper.

Reviewer #3: Please see my specific comments and track-changes in the attached Word document that the authors my consider to enhance clarity.

**Summary and General Comments**

Reviewer #1: The study address an neglected, but important, manifestation of a neglected tropical disease. It is important to include qualitative research methods in the overall assessment of FGS. The study confirms that health education about FGS among health care professionals as well as communities in S.haematobium endemic area is warranted to raise awareness towards essential transmission, prevention and treatment aspects. Most communities are familiar with abbreviation terms such as STIs and HIV related to reproductive health, but not to FGS - not even in S. haematobium endemic areas- across geographical regions, as exemplified in this multi-site study.

Reviewer #2: Overall, this is an informative and important paper which adds to the much-needed evidence base about FGS in the community. It is an impressive sample size for qualitative work and represents a huge amount of time and effort. My comments are minor and suggest areas where additional clarity may help.

Reviewer #3: Mazigo et al present a very nice paper, which highlights very interesting and important aspects regarding community knowledge and perceptions about urogenital schistosomiasis and female genital schistosomiasis. The manuscript is very well structured and written using good quality English throughout. 

The qualitative research study is very well conducted, including a relatively large number of participants from different areas of the United Republic of Tanzania, and differing in sex, age and position in community.

The findings are nicely presented in tables and manuscript text.

My specific comments are provided in the attached Word document and I hope that they can help the authors to further improve their manuscript:

A few important points that I would like to see addressed are:

1) Please indicate the study type in the title

2) How many participants were from Zanzibar and how many from TZ mainland?

3) FGS is the disease, not the infection. This is incorrectly presented several times throughout the manuscript.

4) Mostly, it should read urogenital schistosomiasis and not urinary schistosomiasis, please see my track-changes in the text.

5) The authors may consider and present that in Zanzibar, MDA is directed to schools and communities. Hence, women and adolescent girls are targeted.

6) In Zanzibar, there are few hotspots and many low-prevalences shehias within the districts; the overall prevalence is very low. Hence, I think it is not appropriate to generalize districts as highly endemic. The figures 2 a and 2b in my view are outdated and do not reflect the current situation on the ground. For more up to date information, please see:

Trippler L, Ame SM, Hattendorf J, Juma S, Abubakar S, Ali SM, et al. Impact of seven years of mass drug administration and recrudescence of Schistosoma haematobium infections after one year of treatment gap in Zanzibar: Repeated cross-sectional studies. PLoS Negl Trop Dis. 2021;15(2): e0009127. Epub 2021/02/13.

7) It will be good to describe how the participant sample size was determined. If based on saturation, the process to reach saturation should be described. It will also be good, if the authors can describe on more detail which participants were eligible.

8) Please check whether the data availability statement is in line with PLoS NTD requirements

PLOS authors have the option to publish the peer review history of their article (what does this mean?). If published, this will include your full peer review and any attached files.

Reviewer #1: No

Reviewer #2: No

Reviewer #3: No

Figure Files:

Data Requirements:

Reproducibility:

References

---

## [Editor Report · Decision Letter 1]

3 Sep 2021

Dear Prof. Mazigo,

We are pleased to inform you that your manuscript 'We know about schistosomiasis but we know nothing about FGS”: A qualitative assessment of knowledge gaps about female genital schistosomiasis among communities living in Schistosoma haematobium endemic districts of Zanzibar and Northwestern Tanzania' has been provisionally accepted for publication in PLOS Neglected Tropical Diseases.

Best regards,

Stefanie Knopp

Associate Editor

Justin Remais

Deputy Editor

---

## [Editor Report · Acceptance letter]

28 Sep 2021

Dear Prof. Mazigo,

We are delighted to inform you that your manuscript, "We know about schistosomiasis but we know nothing about FGS”: A qualitative assessment of knowledge gaps about female genital schistosomiasis among communities living in Schistosoma haematobium endemic districts of Zanzibar and Northwestern Tanzania," has been formally accepted for publication in PLOS Neglected Tropical Diseases.

Best regards,

Shaden Kamhawi

co-Editor-in-Chief

Paul Brindley

co-Editor-in-Chief
